# Sustainable Development Goals in Early COVID-19 Prevention and Control

**Taejong Kim [1] and Hyosun Kim [2],***

1   KDI School of Public Policy and Management, Sejong 30149, Korea; tjkim@kdischool.ac.kr
2   Chung-Ang Business School, Chung-Ang University, Seoul 06974, Korea
*   Correspondence: hkim3@cau.ac.kr

**Abstract:** AbstractsRecent failures in COVID-19 prevention and control in some of the richest countries raise questions about the relevance of Sustainable Development Goals (SDGs) in the fight against pandemics. To examine this issue, we adopted the measure of countries' progress for the SDGs in the SDG Index Scores (SDGS) and employed two analytical devices. The first was regression-aided adjustment of the number of deaths and confirmed cases. The second was the use of robust regressions to control the undue influence of outliers. The results are mixed. Between the SDGS and the adjusted infection rates, we found no significant correlation; however, between the SDGS and the adjusted death rates, the correlation was negative and statistically significant. These results provide a nuanced contrast to the hasty conclusions some of us might be tempted to draw from apparent positive correlations between SDGS and the cases and the deaths. The SDGs represent the fruit of painstaking global efforts to encourage and coordinate international action to enhance sustainability. We find the results reassuring, in that they suggest that the countries with higher SDGS have been able to control the devastation of deaths from COVID-19 more effectively, despite being unable to control the propagation of infections.

**Keywords:** COVID-19; Sustainable Development Goals (SDGs); Sustainable Development Goals Scores (SDGS); prevention and control of communicable disease

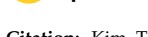



## 1. Introduction

COVID-19 has infected more than 180 million individuals and resulted in close to 4 million deaths across the world as of 30 June 2021 (World Health Organization, WHO henceforth). Economic losses have been hefty, as industries and businesses suffer, small- and medium-sized enterprises in particular [1–4]. Innovative businesses in the SME sector are potentially important change agents for sustainability [5,6], and the blow dealt to the sector by COVID-19 may have long-term negative consequences. Sustainable Development Report (SDR) 2021 lamented setbacks in global progress toward the Sustainable Development Goals (SDGs). Aside from short-lived improvements in SDGs 12–15 (responsible consumption and production; climate action; life below water; and life on land) [7–9], the reversals in the international trends toward the accomplishment of Agenda 2030 have been across-the-board and world-wide [8,10,11]. These setbacks mean heightened vulnerabilities of societies to future threats to global sustainability, including climate change and pandemics similar to COVID-19.

In this paper, we raise and address a related question that might have been considered absurd in the pre-pandemic world: Are SDGs relevant in the fight against global pandemics such as COVID-19? Conspicuous failures in COVID-19 prevention and control in some of the richest countries have rendered this question legitimate. Even though ours is to the best of our knowledge the first investigation to consider the relevance and usefulness of the SDGs in the prevention and control of a global pandemic, similar questions have been addressed elsewhere [11]. It has been widely pointed out that the Global Health Security Index [7], supposedly a measure of country-level preparedness for pandemics, among other

health threats, proved to be a poor predictor of country performances in the fight against COVID-19 [11]. After noting the highly significant positive correlations between COVID-19 infection rates and mortalities on the one hand and the Human Development Index (HDI) scores that tend to be higher in richer countries with well-educated populations and high life expectancy figures on the other, Liu [12] asked whether "extreme" individualism might be responsible for the failures of the richer nations in their fight against the pandemic.

Many other studies have compared national performances in the prevention and control of COVID-19 and examined whether cultural and political factors may have played a role. For instance, Gelfand et al. [13] considered the role of cultural tightness and looseness; Engler et al. [14] looked at the influence of strong protection of democratic principles; and Erman and Medeiros [15] considered uncertainty avoidance and long-term orientation in collective cultural attributes. One may suspect that these studies are partly motivated by the observed failure of the so-called advanced countries to achieve expected health outcomes after the onslaught of the COVID-19 challenge.

Are SDGs an adequate measure of our preparedness and resilience in the face of threats to global sustainability such as COVID-19 pandemic? The SDG Index Scores (SDGS) have been published annually by the Sustainable Development Solutions Network and the Bertelsmann Stiftung to track countries' progress toward the fulfillment of the SDGs in Agenda 2030 [8,16]. Did countries with higher SDGS perform better in the prevention and control of COVID-19? A lot rides on the answer to this question. While the signing of Agenda 2030 in 2015 by 195 member countries of the United Nations was a colossal achievement, if SDGS turns out to be another poor predictor of national performance in COVID-19 challenges, one can hardly doubt the need for revision of at least some of the goals, targets, and indicators in SDGs, given the seriousness of the challenges to sustainability from current and future pandemics.

The main purpose of this brief paper was to carefully assess the question. For the purpose, we employed two analytical devices. The first was regression-aided adjustment of the number of deaths and confirmed cases, incorporating factors contributing to prevalence of communicable diseases and population vulnerability. The second was the use of robust Huber regressions, known to be robust to the presence of outliers [17].

The results are mixed. Between the SDGS and the adjusted infection rates, we found no significant correlation; however, between the SDGS and the adjusted death rates, the correlation was negative and statistically significant. These results provide a nuanced contrast to the hasty conclusions some of us might be tempted to draw from apparent positive correlations between SDGS and the cases and the deaths. The SDGs represent the fruit of painstaking global efforts to coordinate international action to enhance sustainability. We find the results reassuring, in that they suggest that the countries with higher SDGS have been able to control the devastation of deaths from COVID-19 more effectively, despite being unable to control the propagation of infections.

The rest of the paper proceeds as follows. Section 2 presents the motivation of the paper. Section 3 presents the data and the regression results relating the number of COVID-19 deaths and confirmed cases to a range of contributing factors. Section 4 then presents results from the regression of the adjusted death rates and the adjusted confirmed cases rates to the SDGS, with accompanying diagrams, before we offer some concluding remarks in Section 5.

## 2. Background and Hypotheses

### 2.1. Sustainable Development Goals and COVID-19

The Sustainable Development Goals (SDG) represent the fruit of painstaking global efforts to coordinate international action to enhance sustainability. The Sustainable Development Goals consist of 17 goals incorporating 169 targets. The SDG Scores (SDGS) are annually compiled and published by the Sustainable Development Solutions Network and the Bertelsmann Stiftung to track countries' progress in the attainment of the SDGs toward 2030. The SDGS assess each country's overall performance on the SDGs, giving

equal weight to each of the 17 goals. The score measures a country's position between the worst possible outcome (0), and the best (100) or the full achievement of the corresponding goal [8].

Filho et al. [18] suggests COVID-19 represents a serious threat to the attainment of the SDGs. What about the role of SDG in the prevention and control of COVID-19? Does the high level of SDGs protect nations from a pandemic? Conspicuous failures in COVID-19 prevention and control in some of the richest countries have raised the question regarding the relevance of SDGs in the fight against the pandemic. The numbers of COVID-19 deaths and confirmed cases per 100,000 in population tended to be higher in countries with higher scores on SDGs [16]. Figures 1 and 2 show the positive correlations between the SDGS and the number of COVID-19 deaths (Figure 1), and between the SDGS and the number of COVID-19 confirmed cases (Figure 2).

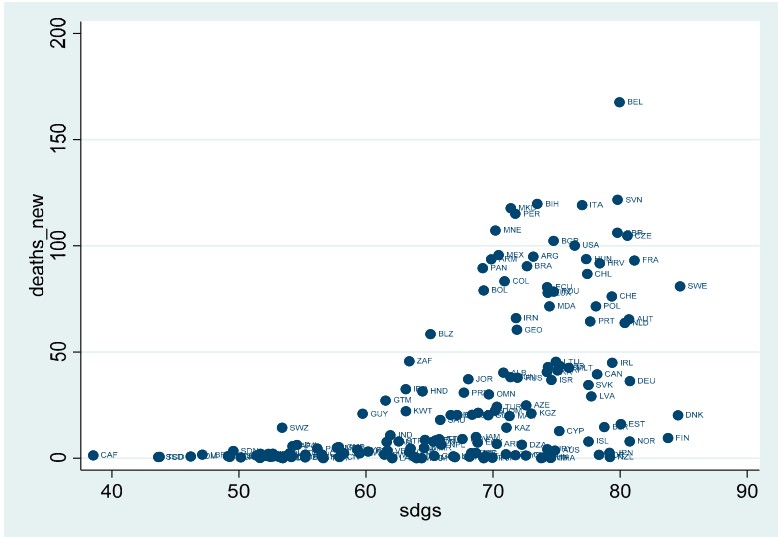

**Figure 1.** Number of COVID-19 deaths and the Sustainable Development Goals Scores (SDGS). Note: The horizontal axis represents the SDGS and the vertical axis the number of deaths due to COVID-19 as of 28 December 2020.

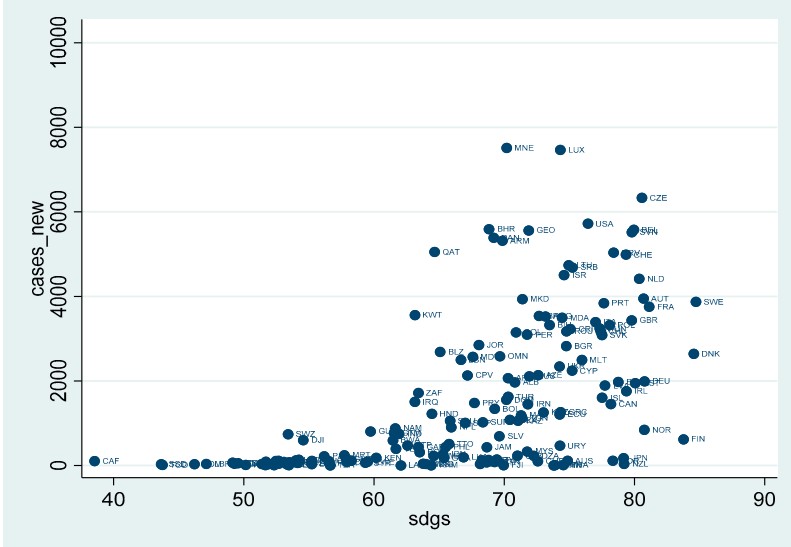

**Figure 2.** Number of COVID-19 confirmed cases and the Sustainable Development Goals. Note: The horizontal axis represents the SDGS and the vertical axis the number of confirmed COVID-19 cases as of 28 December 2020.

We note, however, that these unexpected apparent relationships between SDGs and COVID-19 indicators do not take into account various national differences. De Larochelambert et al. [19] examined the role of demography, public health, economy, politics, and environment on COVID-19 mortality across nations. Moreover, we need to distinguish between the infection rate and the death rate. Even if SDGs may not be effective in the prevention of the disease's spread, SDGs may still contribute to curtailment of deaths from the disease.

*2.2. Hypotheses*

SDGs include good health and well-being or ensuring healthy lives and promoting well-being at all ages (SDG 3). Goal 3 in the SDGs includes the following targets, progress on which should enhance the preparedness of countries to prevent and control a global pandemic.

1.  **SDG 3.3** By 2030, end the epidemics of AIDS, tuberculosis, malaria, and neglected tropical diseases and combat hepatitis, water-borne diseases, and other communicable diseases.
2.  **SDG 3.8** Achieve universal health coverage, including financial risk protection, access to quality essential health-care services, and access to safe, effective, quality, and affordable essential medicines and vaccines for all.
3.  **SDG 3.B** Support the research and development of vaccines and medicines for the communicable and noncommunicable diseases that primarily affect developing countries; provide access to affordable essential medicines and vaccines.
4.  **SDG 3.C** Substantially increase health financing and the recruitment, development, training, and retention of the health workforce in developing countries, especially in least developed countries and small island developing States.
5.  **SDG 3.D** Strengthen the capacity of all countries, in particular developing countries, for early warning, risk reduction, and management of national and global health risks.

Even though some studies point out potential contradictions among items in the SDGs, most assessments find synergies among the elements [20]. Indeed, casual inspection would be sufficient to convince one of the positive reinforcement loops between good health and well-being (SDG 3) and SDGs such as no poverty (SDG 1), zero hunger (SDG 2), quality education (SDG 4), gender equality (SDG 5), and clean water and sanitation (SDG 6), among others.

These observations motivated us to posit the following two hypotheses to examine.

**Hypothesis 1 (H1).** *Higher SDGS are associated with successful prevention of the spread of COVID-19 or lower infection rates.*

**Hypothesis 2 (H2).** *Higher SDGs are associated with successful control of the aftermath of the COVID-19 infections or lower death rates.*

The positive correlations between SDGS and infection/death rates in Figures 1 and 2 cast immediate doubt on the two hypotheses. Thus, we modified the hypotheses in the following way.

**Hypothesis 1′ (H1′).** *Higher SDGS are partially associated with successful prevention of the spread of COVID-19 or lower infection rates, controlling for differences in inherent vulnerability among countries.*

**Hypothesis 2′ (H2′).** *Higher SDGs are partially associated with successful control of the aftermath of the COVID-19 infections or lower death rates, controlling for differences in inherent vulnerability among countries.*

## 3. Data and Methodology

### 3.1. Methodology

This section presents the basic descriptive statistics and preliminary regression results that were necessary to adjust deaths and confirmed cases rates based on underlying susceptibilities and vulnerabilities of the populations in different countries. We first obtained adjusted death rates and adjusted confirmed cases rates through linear regression residuals after controlling for a range of likely contributing factors, and then examined the direction of the correlation between these adjusted rates and the country SDG Scores (SDGS) using robust Huber regressions.

### 3.2. Data

The outcome variables were the number of deaths due to COVID-19 and the number of confirmed COVID-19 cases across countries as of 28 December 2020. The values of these variables were meant to capture the relative success of country public health systems in prevention and control of COVID-19 in the earlier trajectory of the pandemic's propagation, before the arrival of vaccines. We sourced these two variables from the Coronavirus Disease (COVID-19) Dashboard of the World Health Organization [21].

The key right-hand-side variable was the Sustainable Development Goals Scores (SDGS) of each country. The SDGS range theoretically from zero to 100, and higher scores represent further progress to the eventual attainment of the SDGs. We took the values of the SDGS from the Sustainable Development Report 2020 [16].

### 3.3. Adjusted COVID-19 Infection and Death Rates

We considered a range of factors that were meant to capture underlying vulnerabilities of the country populations to a globally communicable disease such as COVID-19. They included the proportion of the elderly in the population (proportion of the population aged 65 and above, %); the proportion of those with diabetes among the adults (%); the proportion of the obese in the population (%); the number of airports; per capita GDP (in USD, PPP); and population density (number of people per square kilometer); and the urbanization rate (the proportion of urban residents, %). The data on the elderly population, the prevalence of diabetes, the prevalence of obesity, per capita GDP, and the urbanization rate were taken from the World Development Indicators [22]. The data on the number of airports was taken from the Airports Council International [23].

Table 1 presents the basic descriptive statistics for the variables analyzed in the paper.

**Table 1.** Basic descriptive statistics.

| Variable | Number of Obs. | Mean | Std. Dev. | Minimum | Maximum |
|---|---|---|---|---|---|
| COVID-19 deaths per 100,000 in population | 190 | 28.59 | 36.90 | 0 | 167.60 |
| COVID-19 confirmed cases per 100,000 in population | 190 | 1653.11 | 1942.43 | 0.57 | 102 |
| Proportion elderly (%) | 175 | 9.27 | 6.51 | 1.16 | 28.00 |
| Proportion diabetic (%) | 186 | 7.88 | 4.24 | 1 | 22.1 |
| Proportion obese (%) | 172 | 18.27 | 8.93 | 2.1 | 37.9 |
| Number of airports | 185 | 222.57 | 1053.83 | 1 | 13513 |
| Population density (per square kilometer) | 185 | 228.65 | 675.08 | 0.14 | 7592.99 |
| GDP per capita (USD, PPP) | 165 | 22,499 | 22,139 | 783 | 121,292 |
| Percentage urban population (%) | 189 | 61.36 | 23.14 | 13.25 | 100 |
| SDGS | 166 | 66.77 | 9.96 | 38.53 | 84.72 |

Table 2 presents results of linear regressions relating the number of confirmed COVID-19 cases per 100,000 in population to a range of likely contributing factors. Among the factors we considered, the proportion of the elderly population, the trade openness, and the proportion of the obese population turned out have statistically significant influences on the number of confirmed cases, as evidenced in the first set of columns (Regression 1). An *F* test of the hypothesis that the remaining factors have no impact failed to reject the null hypothesis. Regression 2 retained only the factors that were identified as having significant impacts. We used the residuals from Regression 2 to produce the adjusted confirmed cases rates for each country.

**Table 2.** Linear regression of COVID-19 confirmed cases on contributing factors.

| Number of COVID-19 Cases per 100,000 | Regression 1 | | | Regression 2 | | |
|---|---|---|---|---|---|---|
| | Coefficient Estimate | Standard Error | *t* | Coefficient Estimate | Standard Error | *t* |
| Proportion elderly (%) | 97.01 | 22.03 | 4.40 | 112.47 | 19.18 | 5.86 |
| Number of airports | 0.28 | 0.28 | 0.99 | | | |
| Per capita GDP (USD, PPP) | 0.01 | 0.008 | 1.39 | | | |
| Proportion diabetic (%) | −25.14 | 34.27 | −0.73 | | | |
| Trade openness | 6.35 | 2.61 | 2.43 | 5.93 | 2.10 | 2.83 |
| Population density (person/km$^2$) | −0.25 | 0.18 | −1.34 | | | |
| Proportion obese (%) | 64.23 | 21.68 | 2.96 | 65.16 | 14.30 | 4.56 |
| Proportion urban (%) | −5.28 | 9.14 | −0.58 | | | |
| Constant | −781.54 | 477.06 | −1.64 | −1160.27 | 304.59 | −3.81 |
| Adj. $R^2$ | 0.47 | | | 0.47 | | |
| Number of obs. | 134 | | | 135 | | |

Table 3 shows results of linear regressions relating the number of COVID-19 deaths per 100,000 in population to a range of likely factors. Among the factors we considered, the proportion of the elderly population, the proportion of the obese population, and the number of airports turned out have statistically significant influences on the number of deaths. See the first set of columns (Regression 1). The F test of the hypothesis that the remaining factors have no impact failed to reject the null hypothesis. Regression 2 retained only the factors that were identified as having significant impacts. We use the residuals from Regression 2 to produce the adjusted death rates for each country.

**Table 3.** Results of linear regressions of COVID-19 deaths on contributing factors.

| Number COVID-19 Deaths per 100,000 in Population | Regression 1 | | | Regression 2 | | |
|---|---|---|---|---|---|---|
| | Coefficient Estimate | Standard Error | *t* | Coefficient Estimate | Standard Error | *t* |
| Proportion elderly (%) | 2.94 | 0.48 | 6.08 | 2.57 | 0.36 | 7.14 |
| Number of airports | 0.019 | 0.006 | 3.07 | 0.005 | 0.002 | 2.24 |
| Per capita GDP (USD, PPP) | −0.0002 | 0.0002 | −1.24 | | | |
| Proportion diabetic (%) | −0.80 | 0.75 | −1.06 | | | |
| Trade openness | 0.07 | 0.06 | 1.31 | | | |
| Population density (person/km$^2$) | −0.003 | 0.004 | −0.79 | | | |
| Proportion obese (%) | 1.18 | 0.48 | 2.47 | 1.02 | 0.27 | 3.83 |
| Proportion urban (%) | −0.01 | 0.20 | −0.05 | | | |
| Constant | −16.47 | 10.48 | −1.57 | −14.98 | 5.23 | -2.87 |
| Adj. $R^2$ | 0.43 | | | 0.40 | | |
| Number of obs. | 134 | | | 167 | | |

Figure 3 uses the residuals from the streamlined regressions (Regression 2) from Tables 2 and 3 to calculate the adjusted infection rate (vertical axis) and the adjusted death rate (horizontal axis), and presents an international scatter plot to compare performance of countries in the prevention and control of the COVID-19 pandemic. Countries in the third quadrant are shown to have outperformed the rest in both prevention and control. We identified 14 countries that registered an adjusted infection rate less than −1000 and an adjusted death rate less than –25: Finland, Australia, Uruguay, Barbados, Norway, Estonia, Latvia, Grenada, Germany, Canada, Thailand, South Korea, Mauritius, and Malta. One might consider these countries as super-performers in the early stage of prevention and control efforts in the COVID-19 fight.

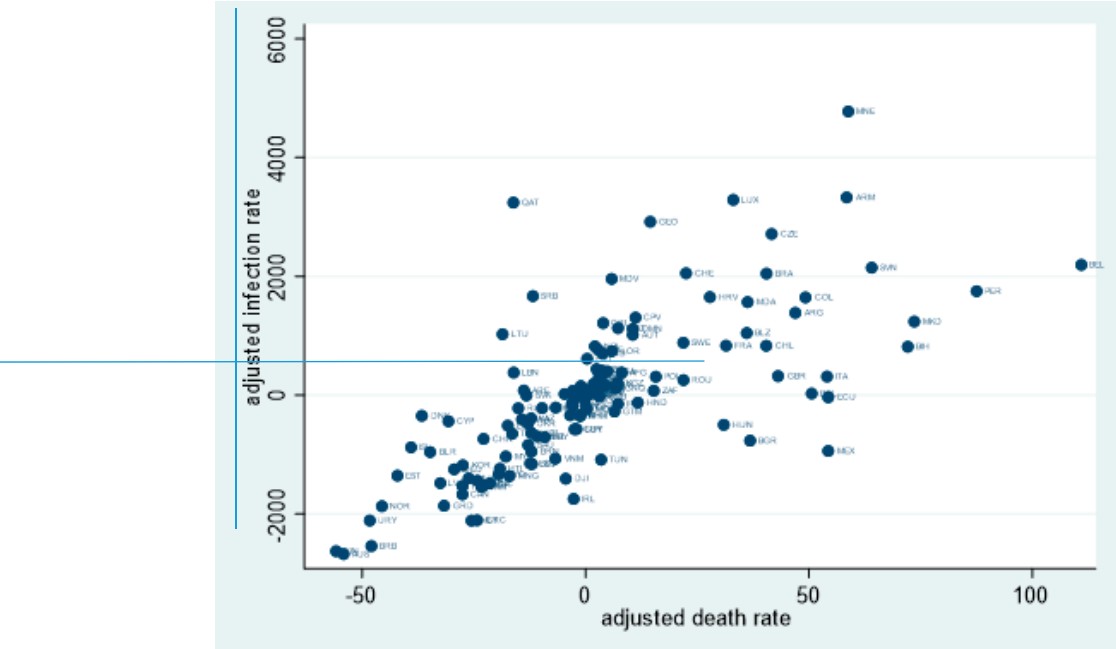

**Figure 3.** International comparison of performance in the prevention and control of COVID-19. Note: The adjusted death rates are residuals from Regression 2 in Table 3; the adjusted infection rates are residuals from Regression 2 in Table 2.

## 4. SDGS and Country Performance in the Prevention and Control of COVID-19

Figures 4 and 5 are scatter plots comparing countries in terms of adjusted COVID-19 infection rate and SDGS (Figure 4) and comparing adjusted COVID-19 death rate and SDGS (Figure 5). Even after adjustments based on regression controls, Figure 4 reveals that SDGS had no apparent systematic impact on the infection rates, except that the variation in the infection rate tended to grow with the SDGS. The pattern that emerges in Figure 5 is perhaps more intriguing. We find a set of countries that seem to form a group of "outliers" (contained in the red ellipsis) that are mostly high-income countries in Europe and middle-income countries in Latin America. If one ignores this group of outliers, the remaining set of countries suggest a fairly strong negative correlation between the SDGS and the adjusted death rate.

Table 4 presents results from robust regressions linking the adjusted infection rates and the adjusted death rates to the SDGS. Robust regression coefficients are results from iterations of Huber regressions after elimination of gross outliers based on the criterion Cook's distance >1, as suggested by Li [24]. Echoing the impressions from the visual inspection, we note that the results diverge depending on the outcome measure. The infection rate, even after regression-aided adjustments and control of undue influences from outliers, does not show any systematic correlation with the SDGS. The p-value from the robust regression of the adjusted death rate on the SDGS is less than 0.08, meaning

that the hypothesis that SDGS have no impact is rejected at 10% level of significance in a two-tailed test and rejected at 5% level of significance in a one-tailed test.

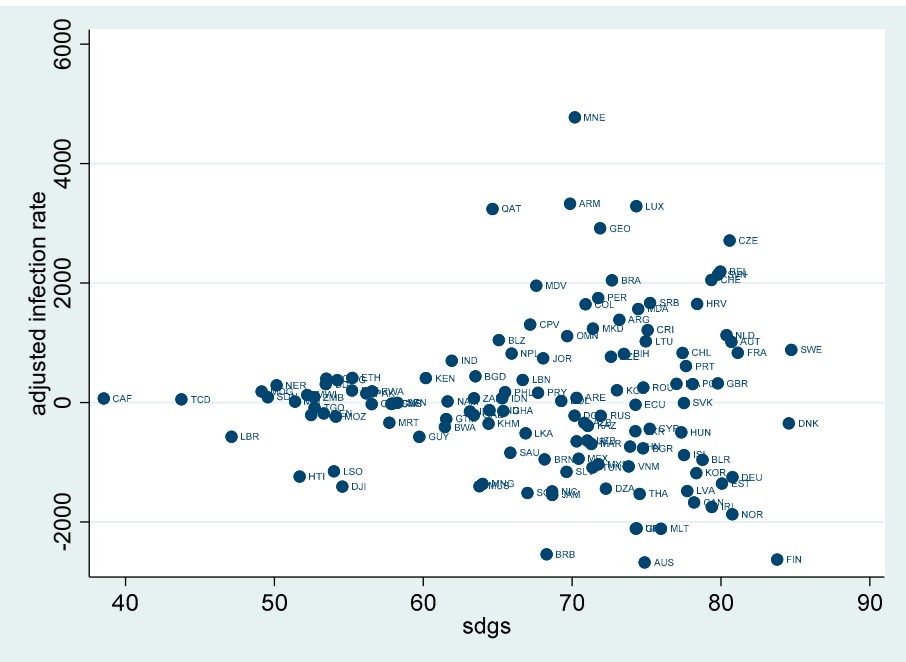

**Figure 4.** Adjusted COVID-19 infection rate vs. SDGS: scatter plot. Note: The horizontal axis measures the SDGS of individual countries and the vertical axis the adjusted infection (confirmed cases) rates, the residuals from Regression 2 in Table 2.

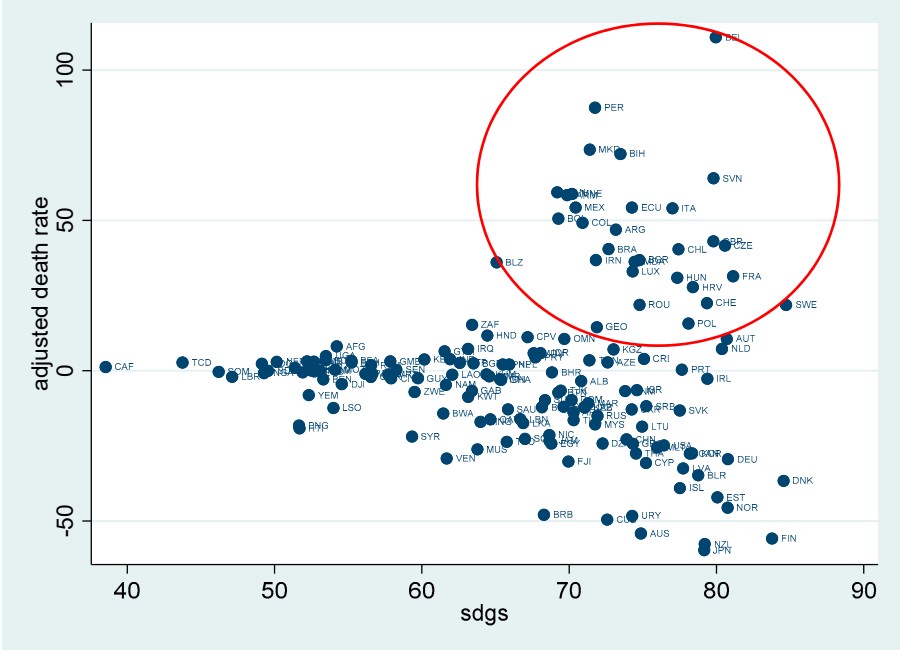

**Figure 5.** Adjusted death rate from COVID-19 vs. SDGS: scatter plot. Note: The horizontal axis measures the SDGS of individual countries and the vertical axis the adjusted infection (confirmed cases) rates, the residuals from Regression 2 in Table 2. The red ellipse gathers together a group of apparent outliers.

**Table 4.** Results of robust regressions associating prevention and control performance and SDGS.

| | Adjusted Infection Rate (Confirmed Cases) | | | Adjusted Death Rate | | |
|---|---|---|---|---|---|---|
| | Estimated Coefficient | Standard Error | *t* | Estimated Coefficient | Standard Error | *t* |
| SDGS | 1.92 | 10.90 | 0.18 | −0.32 | 0.18 | −1.77 |
| Constant | −191.50 | 747.73 | −0.26 | 15.72 | 12.25 | 1.28 |
| Number of obs. | 130 | | | 159 | | |

While we do not show the specific results here, this pattern is observed and emerges in somewhat starker fashion if we limit our attention to an even earlier phase of the pandemic's global propagation: Higher SDGs Scores seem to have been ineffective in the prevention of the pandemic, even after regression controls, although adjusted death rates tended to be lower, the higher the SDGS.

One might wonder what factors possibly explain the presence of outliers in Figure 5. Gelfand et al. [13] suggest a highly plausible candidate factor. They investigated the relationship between cultural tightness-looseness on the one hand and the COVID-19 cases and deaths on the other. Tight cultures have strict norms and impose social sanctions for deviance and may thus be better able to limit cases and deaths, which is indeed what their investigation found.

Gelfand et al. [13] may also provide an explanation about the divergence in the relationships between success in prevention of infection and the SDGS and success in prevention of deaths and the SDGS presented above. Cultural factors may dominate in the determination of success in limiting the number of cases, which relies heavily on adherence to social distancing measures. Countries that perform better in cases of prevention due to their tight cultures may also enjoy advantage in the race to control the number of deaths. On the other hand, SDGS, reflecting the general level of preparedness in terms of resources and technology, may be ineffective in the prevention of case propagation, showing their significance in the limiting of deaths due to COVID-19, in a sense the ultimate performance indicator of the prevention and control efforts by countries.

Earlier in Section 2.2, we noted that SDG 3 targets health and well-being, such as efforts to eliminate major communicable diseases before 2030 (3.3), attainment of universal health coverage (3.8), R&D for vaccines and medicines for communicable and noncommunicable diseases (3.B), health systems upgrades (3.C), and strengthening of the capacity for early earning, risk reduction, and management of national and global health risks (3.D). We also noted the existence of positive feedback loops between SDG 3 and the rest of the SDGs. Perhaps it should thus not surprise us that countries with higher SDGS should be more effective in both mitigation and suppression, the two fundamental strategies in the fight against a pandemic such as COVID-19. Umar et al. [25] evaluated the remarkable progress in the Chinese public health system for SDG 3 during the past two decades and suggests that the vast improvement in the health infrastructure was a critical factor in the mitigation and suppression campaigns against COVID-19 in the country. Across the developing world, huge progress in health and well-being has been accomplished, including in the areas of control of communicable diseases and promotion of maternal and child healthcare. It should be noted, however, that many of the gains have been imperiled by COVID-19 and that the progress had been uneven among and within countries even before the pandemic's onslaught [26,27].

Our finding that countries with high SDGS have, after all, suffered fewer casualties from COVID-19 reconfirms the value of the SDGs as the guideposts for humanity's efforts to enhance sustainability, including our preparedness and resilience in the face of health threats from current and future pandemics. Combined with the observation that progress towards SDGs has witnessed across-the-board setbacks, this means that the global community faces the urgent task of renewing their commitments to the SDGs and redoubling efforts to make progress. This might involve stronger and more effective coordination of

collaboration among the different stakeholder groups, including government, industry, academic institutions, media, and civil society [28–31]. Refinement and stronger application of the principles of knowledge management are called for, in view of the slow-burning nature of pandemics, unlike other disasters such as earthquakes or extreme weather events [32]. Building a better SDGs enterprise going forward largely depends on investment in infrastructure in health, education, environment, and digital connectivity. Thus, serious global efforts must entail expanding the fiscal space for developing countries to finance the requisite investments [7].

## 5. Concluding Remarks

The recent COVID-19 pandemic reminded us that a chain is indeed only as strong as its weakest link. Countries closely affect each other in the dynamics of a pandemic's propagation, and global efforts to fight a pandemic will ultimately succeed only with close global coordination. It is in this vein that this paper investigated the comparative performance of individual countries in their prevention and control efforts.

The investigation in the paper focused on the examination of the hypotheses on whether country preparedness in the form of higher SDGs Scores had explanatory power in the success and failure of the prevention and control of COVID-19. This is an important question to address, as SDGs can be achieved only if they enjoy societal legitimacy, and with the still unfolding challenges from COVID-19, it is hard to think of a more serious issue that might enhance or impair their legitimacy than their relevance in raising our preparedness and resilience in the pandemic context. With efforts to control for differences in underlying susceptibilities and vulnerabilities, and to control for undue influence from outliers, the investigation found that the COVID-19 deaths were effectively reduced by higher SDGS, while the COVID-19 cases were not.

Contrary to possible misinterpretations of apparent positive correlations between SDGS and the cases and deaths, this study confirms that SDGs are indeed relevant even in their guidance of humanity's efforts to fight a global pandemic. We should not forget however that infections that do not lead to deaths still impose significant human costs. Subsequent efforts to better rebuild the public health systems around the world do need to bear these points in mind.

Our findings suggest that rebuilding the momentum for SDG 3 and the other SDGs that are complementary to SDG 3 should be one of the priorities for action to enhance humanity's preparedness to fight COVID-19 and future pandemics. Investing in better public health infrastructure such as the capacity to fight communicable diseases in general (SDG 3.3), universal health coverage (SDG 3.8), R&D for vaccines and medicines for communicable and noncommunicable diseases (3.B), health systems upgrades (3.C), and strengthening the capacity for early learning, risk reduction, and management of national and global health risks (3.D) should receive immediate attention. Discussion also mentioned the need to strengthen coordination among various stakeholder groups for global sustainability, to become more effective and smarter in implementing the strategies, and to buttress financial resources for investment in the developing countries in particular.

**Author Contributions:** Conceptualization, T.K. and H.K.; methodology, T.K.; formal analysis, T.K.; writing—original draft preparation, T.K.; writing—review and editing, H.K. All authors have read and agreed to the published version of the manuscript.

**Funding:** This research received no external funding.

**Data Availability Statement:** We have used the publicly archived datasets for our analysis. The link to the data sources are included in the references.

**Conflicts of Interest:** The authors declare no conflict of interest.

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
