# Peer review of "Sustainable Development Goals in Early COVID-19 Prevention and Control"

_sustainability, doi:10.3390/su13158431_

Round 1

Reviewer 1 Report

  • This paper discusses a very important topic regarding the SDGs agenda. However, there are still several points that need to be considered ;
  • The discussion on the role of each institution is still not explained in detail. On line 29, the authors should explain the role of Penta-Helix or helix collaboration is very important [1–4]. For example: “With the presence of the government, industry, higher education institutions, media, society, and environment that interact with each other. It is necessary to strengthen social capital through Penta helix's synergy”.
  • In the introduction, the author needs to explain what impacts arise as a result of the covid-19 pandemic, for example “economic losses [5–8], crisis on industry sectors and SMEs [9–11] , and pollution reduction [12–14]” . Discuss with detail and specifics in the introduction section. At least the introduction has a minimum of 5 paragraphs.
  • The discussion section needs to be added and extended. In general, this paper fits more into the 'Communication' category than 'article'.

Additional references:

  1. Caraka, R.E.; Noh, M.; Chen, R.C.; Lee, Y.; Gio, P.U.; Pardamean, B. Connecting Climate and Communicable Disease to Penta Helix Using Hierarchical Likelihood Structural Equation Modelling. Symmetry (Basel). 2021, 13, 1–21.
  2. Sudiana, K.; Sule, E.T.; Soemaryani, I.; Yunizar, Y. The development and validation of the Penta Helix construct. Bus. Theory Pract. 2020, 21, 136–145, doi:10.3846/btp.2020.11231.
  3. Martini, L.; Tjakraatmadja, J.H.; Anggoro, Y.; Pritasari, A.; Hutapea, L. Triple Helix Collaboration to Develop Economic Corridors as Knowledge Hub in Indonesia. Procedia - Soc. Behav. Sci. 2012, 52, 130–139, doi:10.1016/j.sbspro.2012.09.449.
  4. Upe, A.; Haluoleo, U.; Sumandiyar, A.; Makassar, U.S.; Jabar, A.; Haluoleo, U. Strengthening of Social Capital through Penta Helix Model in Handling Covid-19 Pandemic. Int. J. Pharm. Res. 2021, 13, doi:10.31838/ijpr/2021.13.01.635.
  5. Hassan, M.K.; Rabbani, M.R.; Ali, M.A.M. Challenges for the Islamic finance and banking in post COVID era and the role of fintech. J. Econ. Coop. Dev. 2020, 41.
  6. Development Bank, A. The Economic Impact of the COVID-19 Outbreak on Developing Asia. 2020, 9, doi:10.22617/BRF200096.
  7. Hudaefi, F.A.; Beik, I.S. Digital zakah campaign in time of Covid-19 pandemic in Indonesia: a netnographic study. J. Islam. Mark. 2021.
  8. Djalante, R.; Shaw, R.; DeWit, A. Building resilience against biological hazards and pandemics: COVID-19 and its implications for the Sendai Framework. Prog. Disaster Sci. 2020, doi:10.1016/j.pdisas.2020.100080.
  9. Lu, L.; Peng, J.; Wu, J.; Lu, Y. Perceived impact of the Covid-19 crisis on SMEs in different industry sectors: Evidence from Sichuan, China. Int. J. Disaster Risk Reduct. 2021, 55, doi:10.1016/j.ijdrr.2021.102085.
  10. Caraka, R.E.; Lee, Y.; Kurniawan, R.; Herliansyah, R.; Kaban, P.A.; Nasution, B.I.; Gio, P.U.; Chen, R.C.; Toharudin, T.; Pardamean, B. Impact of COVID-19 large scale restriction on environment and economy in Indonesia. Glob. J. Environ. Sci. Manag. 2020, 6, 65–84, doi:10.22034/GJESM.2019.06.SI.07.
  11. Belhadi, A.; Kamble, S.; Jabbour, C.J.C.; Gunasekaran, A.; Ndubisi, N.O.; Venkatesh, M. Manufacturing and service supply chain resilience to the COVID-19 outbreak: Lessons learned from the automobile and airline industries. Technol. Forecast. Soc. Change 2021, 163, 120447, doi:10.1016/j.techfore.2020.120447.
  12. Isaifan, R.J. The dramatic impact of coronavirus outbreak on air quality: Has it saved as much as it has killed so far? Glob. J. Environ. Sci. Manag. 2020, doi:10.22034/gjesm.2020.03.01.
  13. Ogen, Y. Assessing nitrogen dioxide (NO2) levels as a contributing factor to coronavirus (COVID-19) fatality. Sci. Total Environ. 2020, 726, doi:10.1016/j.scitotenv.2020.138605.
  14. Čurović, L.; Jeram, S.; Murovec, J.; Novaković, T.; Rupnik, K.; Prezelj, J. Impact of COVID-19 on environmental noise emitted from the port. Sci. Total Environ. 2021, 756, doi:10.1016/j.scitotenv.2020.144147.

Author Response

Dear Sir/Madam, 
My author and I feel deeply indebted to your generous and constructive comments and suggestions, and we are grateful for the chance to revise and resubmit our manuscript. 
More than anything else, your review led us to think more carefully and systematically about the context of our investigation, including the relevant literature streams, the institutional backgrounds, and the on-going international discourse regarding the pandemic and the Sustainable Development Goals. We also felt the need to dig deeper about policy implications from our findings. The kind list of references you gave us proved to be a tremendous help in the process. 
We have done our best to incorporate your valuable suggestions in revision. As a result, we believe the revised manuscript is a huge improvement compared to the initial submission. Down below, we are providing explanations of our point-by-point responses to your comments and suggestions. The revised manuscript also contains memos to describe the changes in the text, and where relevant, we noted in the memos the Reviewers’ comments and suggestions that had motivated the changes, including yours. We humbly hope that you will kindly note the improvements and see that the paper feels now more like a proper academic article. 
Studying the reviewers’ notes including yours and revising our manuscript have been a rewarding learning process for us. In case you have further points that need our attention, we will be more than pleased to engage in another round of revisions to act upon your suggestions. 
Thanks once again. 
Sincerely yours,
Hyosun on behalf of the team of co-authors 

Hyosun Kim, Professor, Chung Ang University

[1]. This paper discusses a very important topic regarding the SDGs agenda.
However, there are still several points that need to be considered ;

Thanks a lot for the encouraging assessment. We are aware of the weaknesses that the initial submission included. We have done our best to respond to your suggestions and to improve the manuscript in other ways as well. 

[2]. The discussion on the role of each institution is still not explained in detail.
On line 29, the authors should explain the role of Penta-Helix or helix collaboration is very important [1–4]. For example: “With the presence of the government, industry, higher education institutions, media, society, and environment that interact with each other. It is necessary to strengthen social capital through Penta helix's synergy”.

Thanks a lot. My coauthor and I were not previously aware of the literature on the Penta Helix and Triple Helix collaboration. Your suggestion made us see the need to more fully develop practical policy implications from our findings. As we are painfully aware, COVID-19 dealt setbacks to the positive international trends for the SDGs. A key finding from our paper is the significant relevance of the SDGs in the humanity’s fight against a pandemic such as COVID-19. In combination, they imply an urgent need for the international community to renew the commitments for the SDGs and strengthen and smarten our efforts to build forward better. In the new paragraph that we added to Section 4, we dwell on practical implications from our findings, and emphasize the need for more effective coordination of collaboration among various stakeholder groups including the government, the industry, the civil society, the media, and the academia. We cited all four references you kindly gave us in the discussion. Thanks once again.  

[3]. In the introduction, the author needs to explain what impacts arise as a result of the covid-19 pandemic, for example “economic losses [5–8], crisis on industry sectors and SMEs [9–11] , and pollution reduction [12–14]” .
Discuss with detail and specifics in the introduction section. At least the introduction has a minimum of 5 paragraphs.

Thanks for the excellent suggestions. We added a new paragraph at the beginning of the introductory section to specifically provide a discussion on the impacts of COVID-19 in various domains. We cited almost all of your suggested references in the expanded discussion. More broadly, we realized that we need to provide the context and backgrounds of the investigation, in terms of the streams of relevant literature, institutional backgrounds, and on-going international discourse on the COVID-19 pandemic and SDGs. For that, we expanded the initial manuscript’s single paragraph presenting the main research question into a set of three related paragraphs.   

[4]. The discussion section needs to be added and extended. In general, this paper fits more into the 'Communication' category than 'article'.

As we explained in our response to your comment [2], we added a new paragraph in Section 4 to provide more discussion of practical and theoretical implications of our findings. As well, we also added a whole new subsection (2.2) to more carefully and systematically develop and present our main hypotheses. 

Additional references: 
The list of additional references was very helpful in our efforts to revise the manuscript and incorporate your constructive suggestions. I already noted how we used and cited some of the references in the revision. Here let me explain why we decided not to cite the limited number of references highlighted in orange in the revision. While we learned a lot from them, some of them (5, 7, and 10) are rather narrowly focused on a particular country or a particular religious practice, and two of them (8 and 13) do not directly impact on the development of the narrative and the exposition in our paper.  

Caraka, R.E.; Noh, M.; Chen, R.C.; Lee, Y.; Gio, P.U.; Pardamean, B. Connecting Climate and Communicable Disease to Penta Helix Using Hierarchical Likelihood Structural Equation Modelling. Symmetry (Basel). 2021, 13, 1–21.

Sudiana, K.; Sule, E.T.; Soemaryani, I.; Yunizar, Y. The development and validation of the Penta Helix construct. Bus. Theory Pract. 2020, 21, 136–145, doi:10.3846/btp.2020.11231.
3. Martini, L.; Tjakraatmadja, J.H.; Anggoro, Y.; Pritasari, A.; Hutapea, L. Triple Helix Collaboration to Develop Economic Corridors as Knowledge Hub in Indonesia. Procedia - Soc. Behav. Sci. 2012, 52, 130–139, doi:10.1016/j.sbspro.2012.09.449. 
4. Upe, A.; Haluoleo, U.; Sumandiyar, A.; Makassar, U.S.; Jabar, A.; Haluoleo, U. Strengthening of Social Capital through Penta Helix Model in Handling Covid-19 Pandemic. Int. J. Pharm. Res. 2021, 13, doi:10.31838/ijpr/2021.13.01.635.

5. Hassan, M.K.; Rabbani, M.R.; Ali, M.A.M. Challenges for the Islamic finance and banking in post COVID era and the role of fintech. J. Econ. Coop. Dev. 2020, 41.
6. Development Bank, A. The Economic Impact of the COVID-19 Outbreak on Developing Asia. 2020, 9, doi:10.22617/BRF200096.
7. Hudaefi, F.A.; Beik, I.S. Digital zakah campaign in time of Covid-19 pandemic in Indonesia: a netnographic study. J. Islam. Mark. 2021.

8. Djalante, R.; Shaw, R.; DeWit, A. Building resilience against biological hazards and pandemics: COVID-19 and its implications for the Sendai Framework. Prog. Disaster Sci. 2020, doi:10.1016/j.pdisas.2020.100080.

9. Lu, L.; Peng, J.; Wu, J.; Lu, Y. Perceived impact of the Covid-19 crisis on SMEs in different industry sectors: Evidence from Sichuan, China. Int. J. Disaster Risk Reduct. 2021, 55, doi:10.1016/j.ijdrr.2021.102085.

10. Caraka, R.E.; Lee, Y.; Kurniawan, R.; Herliansyah, R.; Kaban, P.A.; Nasution, B.I.; Gio, P.U.; Chen, R.C.; Toharudin, T.; Pardamean, B. Impact of COVID-19 large scale restriction on environment and economy in Indonesia. Glob. J. Environ. Sci. Manag. 2020, 6, 65–84, doi:10.22034/GJESM.2019.06.SI.07.

11. Belhadi, A.; Kamble, S.; Jabbour, C.J.C.; Gunasekaran, A.; Ndubisi, N.O.; Venkatesh, M. Manufacturing and service supply chain resilience to the COVID-19 outbreak: Lessons learned from the automobile and airline industries. Technol. Forecast. Soc. Change 2021, 163, 120447, doi:10.1016/j.techfore.2020.120447.
12. Isaifan, R.J. The dramatic impact of coronavirus outbreak on air quality: Has it saved as much as it has killed so far? Glob. J. Environ. Sci. Manag. 2020, doi:10.22034/gjesm.2020.03.01.
13. Ogen, Y. Assessing nitrogen dioxide (NO2) levels as a contributing factor to coronavirus (COVID-19) fatality. Sci. Total Environ. 2020, 726, doi:10.1016/j.scitotenv.2020.138605.
14. Čurović, L.; Jeram, S.; Murovec, J.; Novaković, T.; Rupnik, K.; Prezelj, J. Impact of COVID-19 on environmental noise emitted from the port. Sci. Total Environ. 2021, 756, doi:10.1016/j.scitotenv.2020.144147.

Reviewer 2 Report

This "fast" paper is interesting and contents are consistent with the aim. Anyway managing a pandemic is a complex task. The background is too brief and concepts are not linked to literature streams. It's not clear how the results can be exploited to drive research and support health organisations, and policymakers, to face a COVID-19 like pandemic. Authors need to reinforce background and conclusion sections with references to sustainable development papers and knowledge management during pandemics (e.g.: Ammirato S., Linzalone R., Felicetti A.M., 2020, "Knowledge management in pandemics. A critical literature review", Knowledge Management Research & Practice)

Author Response

Dear Sir/Madam, 
My author and I feel deeply indebted to your generous and constructive comments and suggestions, and we are grateful for the chance to revise and resubmit our manuscript. 
More than anything else, your review led us to think more carefully and systematically about the context of our investigation, including the relevant literature streams, the institutional backgrounds, and the on-going international discourse regarding the pandemic and the Sustainable Development Goals. We also felt the need to dig deeper about policy implications from our findings. The reference you gave us proved to be a valuable help in the process. 
We have done our best to incorporate your valuable suggestions in revision. As a result, we believe the revised manuscript is a huge improvement compared to the initial submission. Down below, we are providing explanations of our point-by-point responses to your comments and suggestions. The revised manuscript also contains memos to describe the changes in the text, and where relevant, we noted in the memos the Reviewers’ comments and suggestions that had motivated the changes, including yours. We humbly hope that you will kindly note the improvements and see that the paper feels now more like a proper academic article, and not so “fast”. 
Studying the reviewers’ notes including yours and revising our manuscript have been a rewarding learning process for us. In case you have further points that need our attention, we will be more than pleased to engage in another round of revisions to act upon your suggestions. 
Thanks once again. 
Sincerely yours,
Hyosun
On behalf of the team of co-authors 

Hyosun Kim, Professor, Chung Ang University 

Comments and Suggestions for Authors
This "fast" paper is interesting and contents are consistent with the aim. 
Thanks a lot for the encouraging assessment. We are aware of the weaknesses that the initial submission included. We have done our best to respond to your suggestions and to improve the manuscript in other ways as well. 
Anyway managing a pandemic is a complex task. The background is too brief and concepts are not linked to literature streams. 
Thanks for the thoughtful assessments. We added a new paragraph at the beginning of the introductory section to specifically provide a discussion on the impacts of COVID-19 in various domains. More broadly, we realized that we need to provide the context and backgrounds of the investigation, in terms of the streams of relevant literature, institutional backgrounds, and on-going international discourse on the COVID-19 pandemic and SDGs. For that, we expanded the initial manuscript’s single paragraph presenting the main research question into a set of three related paragraphs. Our efforts to revise the paper in this regard also enabled us to see connections to streams of literature that had previously evaded our attention. Thanks once again.    
It's not clear how the results can be exploited to drive research and support health organisations, and policymakers, to face a COVID-19 like pandemic. Authors need to reinforce background and conclusion sections with references to sustainable development papers and knowledge management during pandemics (e.g.: Ammirato S., Linzalone R., Felicetti A.M., 2020, "Knowledge management in pandemics. A critical literature review", Knowledge Management Research & Practice)
Thanks a lot for the kind and thoughtful suggestions. Your suggestions made us see the need to more fully develop practical policy implications from our findings. As we are painfully aware, COVID-19 dealt setbacks to the positive international trends for the SDGs. A key finding from our paper is the significant relevance of the SDGs in the humanity’s fight against a pandemic such as COVID-19. In combination, they imply an urgent need for the international community to renew the commitments to the SDGs and strengthen and smarten our efforts to build forward better. In the new paragraph that we added to Section 4, we dwell on practical implications from our findings, and emphasize the need for more disciplined and systematic applications of lessons from the knowledge management perspective, among other actionable recommendations. We duly cited the reference your kindly gave us. 

Submission Date
01 July 2021
Date of this review
04 Jul 2021 16:46:39

Round 2

Reviewer 1 Report

The author did the revision very well. However, the reference used is still inadequate.

I give a recommendation for this paper to be published after completing several points.

Author Response

Thank you very much for your thoughtful comments.

We benefitted a lot from your review. 
Please find the attached file for the detailed revision. 

Best, 
Authors.

Reviewer 2 Report

All the suggestions have been taken into account.

Well done.

Author Response

Thank you very much for warm encouragement. 
We benefit a lot from your thoughtful comment.